# SARS-CoV-2 Vaccines: The Advantage of Mucosal Vaccine Delivery and Local Immunity

**DOI:** 10.3390/vaccines12070795

**Published:** 2024-07-18

**Authors:** Joshua Tobias, Peter Steinberger, Joy Wilkinson, Gloria Klais, Michael Kundi, Ursula Wiedermann

**Affiliations:** 1Institute of Specific Prophylaxis and Tropical Medicine, Center for Pathophysiology, Infectiology and Immunology, Medical University of Vienna, 1090 Vienna, Austria; 2Division of Immune Receptors and T Cell Activation, Institute of Immunology, Center for Pathophysiology, Infectiology and Immunology, Medical University of Vienna, 1090 Vienna, Austria; peter.steinberger@meduniwien.ac.at; 3Department of Environmental Health, Center for Public Health, Medical University of Vienna, 1090 Vienna, Austria; michael.kundi@meduniwien.ac.at

**Keywords:** respiratory infections, parenteral vaccines, nasal vaccines, mucosal immunity, SARS-CoV-2, B-cell peptide-based vaccine, prime-boost immunization

## Abstract

Immunity against respiratory pathogens is often short-term, and, consequently, there is an unmet need for the effective prevention of such infections. One such infectious disease is coronavirus disease 19 (COVID-19), which is caused by the novel Beta coronavirus SARS-CoV-2 that emerged around the end of 2019. The World Health Organization declared the illness a pandemic on 11 March 2020, and since then it has killed or sickened millions of people globally. The development of COVID-19 systemic vaccines, which impressively led to a significant reduction in disease severity, hospitalization, and mortality, contained the pandemic’s expansion. However, these vaccines have not been able to stop the virus from spreading because of the restricted development of mucosal immunity. As a result, breakthrough infections have frequently occurred, and new strains of the virus have been emerging. Furthermore, SARS-CoV-2 will likely continue to circulate and, like the influenza virus, co-exist with humans. The upper respiratory tract and nasal cavity are the primary sites of SARS-CoV-2 infection and, thus, a mucosal/nasal vaccination to induce a mucosal response and stop the virus’ transmission is warranted. In this review, we present the status of the systemic vaccines, both the approved mucosal vaccines and those under evaluation in clinical trials. Furthermore, we present our approach of a B-cell peptide-based vaccination applied by a prime-boost schedule to elicit both systemic and mucosal immunity.

## 1. Respiratory Tract Infections and Mucosal Immune Responses

Respiratory infections are a leading cause of illness and mortality worldwide [1,2]. The upper and the lower respiratory tract (URT and LRT, respectively) comprise the respiratory system. Generally, the upper respiratory tract encompasses the nasal cavity and pharynx, and the LRT includes the trachea, bronchi, and lungs [3]. The URT is considered as the primary site for infection with respiratory pathogens, including the severe acute respiratory syndrome coronavirus-2 (SARS-CoV-2).

The respiratory mucosa, covering the respiratory epithelium, consists of a thick layer of protective mucus, which confers mucosal immunity by orchestrated innate and adaptive responses against viral respiratory infections [3,4]. Primary infections, which lead to acute immune responses, are mediated by the innate immune system and the subsequent activation of adaptive immunity [5,6]. The predominant antibody isotype in secretions and the upper respiratory system is immunoglobulin A (IgA). Although also detected in serum, the mucosal/secretory IgA (s-IgA) is attributed to the mucosal sites and its concentration exceeds by 2.5-fold the concentration of IgG, which is abundant in serum [7]. In serum, IgA is present in a monomeric form, while the s-IgA is in a polymeric, particularly dimeric or tetrameric, form setup [8,9]. The lack of a secretory component in serum IgA enables the antibodies to bind to Fcα-receptor I (FcαRI) expressed by myeloid cells, such as monocytes, neutrophils, and some subsets of macrophages and dendritic cells (DCs), resulting in the induction of pro-inflammatory responses [10]. 

Macrophages and natural killer (NK) cells are among the vital immune cells in the URT mucus, and they play a key role in the innate immune response at this site. In addition, the presence of defensive compounds such as lysozyme and lactoferrin enhances the innate immune response [3,11]. The adaptive immune response is coordinated by antigen-specific cytotoxic T lymphocytes, which kill infected host cells. Antigen-specific antibodies produced by B cells have the capacity to eliminate the infected cells and neutralize the pathogen [3].

## 2. Mucosal Versus Systemic Vaccines

Vaccines are among the prominent medical inventions in human history, with the proven capacity to save countless lives worldwide [12,13]. Respiratory droplets carrying infectious agents, for example the influenza virus, which can be transferred to other individuals, are the most effective route of transmission, infection initiation, and, subsequently, upper respiratory symptomatic disease manifestation in the infected individual [14]. Thus, disruption of the spread of a virus is highly significant for reducing the disease’s transmission and its impact [15]. 

The generation of virus-specific IgA antibodies at the nasal surface is the primary defense mechanism against viral infections in the upper respiratory tract [16,17]. As shown in Figure 1, intranasal vaccinations are thought to offer two extra levels of protection in comparison to parenteral vaccinations: s-IgA and resident memory B and T cells in the respiratory mucosa [18,19,20]. On the other hand, parenteral vaccinations generally cause a greater induction of virus-specific IgG in the blood than nasal vaccinations [16,21,22].

Primarily, the s-IgA antibodies serve as neutralizing antibodies (nAbs) to inhibit viral or bacterial binding to the mucosa and, thus, protect it from the entry of the pathogen to cells [8,23,24]. Bellanti et al. were among the first who characterized the antibodies that have a viral neutralization capacity and showed the importance of sIgA as a predictive marker of immunity against respiratory infections [25,26,27,28]. Furthermore, due to their lack of capacity to activate the classical complement pathway, unlike IgG antibodies, the immune response mediated by s-IgA antibodies at the mucosal site is non-inflammatory [29,30,31,32]. The parenteral administration of vaccines has been a highly effective strategy for preventing and controlling infectious diseases. Parenteral vaccines are administered through intramuscular, subcutaneous, or intradermal injections, offering a rapid and efficient delivery of antigens to the immune system, and allowing the induction of robust immune responses [33,34]. On the other hand, such vaccines induce low mucosal IgA and IgG antibodies in the URT and LRT [33]. Although serum IgG antibodies can diffuse to mucosal surfaces by passive diffusion across the epithelium [35], booster vaccinations are required to overcome the waning of the induced circulating antibodies [33]. Earlier studies have shown that the parenteral and mucosal immunization of naïve individuals against novel antigens induces the respective type of immune response, but not the reciprocal type of response, suggesting a substantial mutual independence of systemic and mucosal responses [36,37].

Mucosal vaccines can lessen viral shedding and transmission as well as prevent viral replication at the vaccination site in the event of re-infection by inducing the production of resident memory B and T cells, which have already encountered the virus and can respond more quickly than systemic memory cells [18]. An additional advantage of the mucosal route of vaccination compared to the injectable/systemic route lies in the better homing and formation of specific tissue resident memory CD4 T cells and CD8 T cells, as shown for SARS-CoV-2 [38,39,40]. 

A crucial role of virus-specific T-cell production within mucosal tissues in mediating the host response against chronic viral infections and the resulting immunopathology has been well documented [41,42,43]. This indicates the superiority of the mucosal route of vaccination, as it not only prevents establishment of the infection but also promotes the control of chronic viral infections [44,45].

One of the most extensively studied respiratory viruses is the influenza virus, against which several types of vaccines have been licensed, such as the inactivated influenza vaccines (IIV), which are injected, and recommended for all children 6 months of age and older, and the intranasal live attenuated influenza vaccine (LAIV) for healthy older children aged 2 years and above [46,47]. While strain-specific antibody-mediated protection is achieved by IIV, the vaccine does not induce a long-term cellular response. On the other hand, broader humoral and cellular immune responses, including additional mucosal IgA antibodies, that protect the site of viral entry against recurrent infection have been found to be elicited by LAIV [46,47,48,49,50,51,52]. Furthermore, no adjuvant is needed for LAIV, whereas IIV often requires an adjuvant [50]. In this regard, in earlier investigations, the intranasal administration of an inactivated influenza virosome-based vaccine, together with the mutated but enzymatically active heat labile toxin (LT) of Escherichia coli, was reported to be strongly associated with the development of facial nerve paralysis (Bell’s Palsy) [53,54]. An important aspect of mucosal vaccination is the role or presence of the pre-existing immunity to the pathogen. For example, pre-existing antibodies derived from seasonal influenza infections or inactivated influenza vaccines can impair LAIV’s effectiveness and subsequently interject the establishment of immunity at the mucosal site [55,56]. In line with this, a higher-fold induction of serum hemagglutination-inhibition antibodies, associated with protection from influenza infection, has been demonstrated among 2–8-year-old children and seronegative adults vaccinated with LAIV [57].

The characteristics of the immune responses following mucosal and parenteral vaccinations point to a vaccination strategy combining the advantages of both vaccination routes. The following sections of this review further discuss parenteral and mucosal vaccinations, with a focus on SARS-CoV-2. 

## 3. SARS-CoV-2

On 8 December 2019, a novel coronavirus (CoV) case in the Chinese province of Hubei was recorded [58]. Within a very short time, on 9 January 2020, the complete genome sequence of the virus was released, leading to its identification as a novel coronavirus, with the genomic sequence being related to the previously described severe acute respiratory syndrome coronavirus (SARS-CoV); therefore, it was named SARS-CoV-2 [58,59,60]. The acute respiratory disease linked to the virus is known as coronavirus disease 19 (COVID-19) (WHO-situation report-22). On 11 March 2020, the World Health Organization (WHO) declared the SARS-CoV-2 pandemic. Since its emergence, until mid-June 2024, the virus has led to 704,753,890 confirmed cases and 7,010,681 fatalities worldwide (https://www.worldometers.info/coronavirus/ (accessed on 12 June 2024)), and has caused an unprecedented burden on national health systems, economies, and general human welfare globally [61].

### 3.1. Pathogenicity of SARS-CoV-2

Ever since the emergence of SARS-CoV-2, a series of COVID-19 outbreak waves has struck the world due to evolving mutants from the original SARS-CoV-2 strain. The illness caused by COVID-19 ranges from asymptomatic to critically symptomatic clinical manifestations (severe acute respiratory distress, pneumonia, damage to several organs, and even death) [62,63]. Reports during the initial phase of the pandemic indicated that up to 20 percent of those with COVID-19 developed severe disease and required hospitalization, among which up to one-quarter needed intensive care unit admission [64,65,66]. The severe type of COVID-19 mostly impacts those who are immunosuppressed, old, or have comorbidities [67,68]. A surge and hyper-induction of pro-inflammatory cytokines, also known as a “cytokine storm,” characterized by the uncontrolled and elevated release of cytokines including interleukin (IL)-1, IL-6, TNF-α, and interferons along with low Treg levels, was the key clinical aspect of the earlier variants of SARS-CoV-2 [69,70,71]. 

In addition to a very high death toll of the pandemic across the globe, increasing attention has been drawn to the prolonged or late-onset sequelae of SARS-CoV-2 infection, colloquially referred to as ‘long-COVID-19′ syndrome [72], which affects, e.g., neurological, respiratory, cardiovascular, gastrointestinal, renal, immunological, and reproductive organs [73,74,75]. SARS-CoV-2 is extensively evidenced to cause many neurological diseases similar to neurological manifestations previously reported for other respiratory viral infections, referred to as post-viral infection syndrome [76]. However, the neurological symptoms of COVID-19 are highly frequent and disabling [77]. Neurological complications include a long-term presence of the symptoms, such as headaches, insomnia, depression, anxiety, dizziness, seizures, and mood swings [78,79,80,81,82,83], and such complications may be exacerbated either during the acute SARS-CoV-2 infection or during its post-acute phase [84,85,86]. Critical illness polyneuropathies and critical illness myopathies are important neurological complications in critically ill patients with COVID-19 [87]. In early investigations of COVID-19-positive patients in Wuhan, it was demonstrated that 36.4% of patients displayed neurological manifestations and 8.9% presented peripheral nervous system symptoms, the most prevalent of which was anosmia (5.1%) [88].

### 3.2. Virulence Factors of SARS-CoV-2

The sequenced genome of SARS-CoV-2 was the basis for the understanding of its viral pathogenicity and the development of therapeutics and vaccines for the virus. Two-thirds of the large viral genome (>30 kb) typically encodes the replicase, and the remaining genome encodes the structural and accessory proteins. The viral particle is composed of a helical nucleocapsid (N) structure created by an association of phosphoproteins and genomic RNA, which is enclosed by a lipid bilayer in which the spike (S), the membrane (M) and the envelope (E) structural proteins are anchored [89,90].

A wide range of cell types can be infected by SARS-CoV-2 including alveolar cells, macrophages, endothelial cells, kidney cells, intestinal epithelial cells, monocytes, neurons, glial cells, and neuroepithelial cells [91,92,93,94]. Among the structural proteins of SARS-CoV-2, the S protein plays an essential role in viral attachment, fusion, entry, and transmission [90,95,96]. The protein comprises an *N*-terminal S1 subunit responsible for the virus–receptor binding, and a *C*-terminal S2 subunit responsible for virus–cell membrane fusion. The S1 is further divided into an *N*-terminal domain (NTD) and a receptor-binding domain (RBD). The RBD, spanning the amino acid (AA) positions 319–541, includes the receptor-binding motif (RBM; AAs 437-508) [96] that contains the majority of the binding epitopes involved in the SARS-CoV-2 and ACE2 interaction [90].

The initial interaction between the virus and the human cells is via the RBD, which binds to the ACE2 [97]. In addition to its role in the pathogenicity of the virus, the ACE2 possesses numerous physiological functions, including protection against lung injury [98]. 

The binding of the RBD to the ACE2 leads to the disassociation of the S1 with the ACE2 receptor, which subsequently prompts the shifting of the S2 subunit from pre-fusion to a more stable post-fusion state that consequently drives the viral fusion with the host cell [99]. The fusion and entry of SARS-CoV-2, driven by the S2 subunit, requires the priming of the S protein by cellular proteases which cleave the protein at the S1/S2 and the S2′ site and allow the fusion of viral and cellular membranes [100]. The S protein contains two cleavage sites for cellular proteases, i.e., S1/S2 between the two subunits S1 and S2, and S2′ in the S2 subunit, with the latter being essential for inducing membrane fusion. It was reported that the S1/S2 cleavage alone does not trigger membrane fusion [97,101]. 

### 3.3. SARS-CoV-2 Variants

SARS-CoV-2, like other coronaviruses, has a high mutation rate and, since the detection of the original strain, several variants of the virus have been evolved [102,103,104]. Variants of SARS-CoV-2 have been categorized by the World Health Organization (WHO) into the categories variations under monitoring (VUMs), variants of interest (VOIs), and variants of concern (VOCs). VOCs were divided into five groups: Alpha (B.1.1.7), Beta (B.1.351), Gamma (P.1), Delta (B.1.617.2), and Omicron (B.1.1.529) variants [105]. Due to its high transmissibility [106] and immune evasion potency [107], the Omicron variant became predominant and outcompeted the previous variants, although the disappearance of the pre-Omicron variants may have been attributed to the development of immunity induced due to infection with the previous variants [108]. Following Omicron, many Omicron sub-lineages, such as the recent variants XBB.1.5 (Kraken), XBB.1.16 (Arcturus), EG.5.1 (Eris), BQ.1, and BQ.1.1, have emerged [105,109].

The variant BA.2.86, referred to as the “second generation of BA.2”, was detected in late 2023 in several countries [110]. However the BA.2.86 variant did not become predominant, due to its relative sensitivity to nAbs compared to XBB variants [111], and, instead, a descendent of the BA.2/BA.2.86 variants, JN.1, is currently the predominant subvariant [111].

### 3.4. Immunity against SARS-CoV-2

Shortly after SARS-CoV-2 infection, the vast majority of individuals develop an antibody response, mainly directed against the highly immunogenic epitopes on the S1 and S2 domains of the S protein [112,113]. NAbs, providing robust protection against subsequent reinfection with the same strain [114], are commonly directed to the RBM [115,116], and to a lesser extent to the *N*-terminal domain of the S [117] and N [118] proteins. We have, earlier, shown that, among individuals infected with SARS-CoV-2, regardless of the disease severity, RBD-specific antibodies and nAbs can be detected for at least six months following infection [119]. NAbs are a crucial predictor of survival in COVID-19 patients [112,113]. Studies have shown a lower risk of death from COVID-19 in immunized/infected individuals with high-titer convalescent-phase plasma, who consequently exhibited a substantial viral load reduction [117,120,121,122,123]; this points to the role of cell-mediated immunity (primarily T-cell response) in controlling SARS-CoV-2 and the disease’s severity [124].

## 4. Treatments against SARS-CoV-2 by Monoclonal Antibodies (mAbs) and Vaccination

### 4.1. Therapeutic mAbs

One approach to pre- and post-exposure prophylaxis or treatment is passive antibody application [125,126]. Using this over-a-century-old strategy, nAbs are isolated from recovered individuals, i.e., convalescent antibody treatment, as they have been shown to play a crucial role in combating viral infection and severe disease manifestation [125,127]. The impact of the COVID-19 pandemic, particularly during the early stages of the outbreak, led to extensive efforts to develop mAbs, as either single agents or cocktails, with the strong capacity to target the original SARS-CoV-2 S protein [102,128,129]. Such mAbs have been used for therapeutic treatment and the pre-exposure or post-exposure prophylactic treatment of mild-to-moderate COVID-19 (Table 1) [102,130,131,132]. Despite the exceptional therapeutic effect of the mAbs, with the continuing emergence of SARS-CoV-2 variants, most mAbs, due to a reduced/lack of binding capacity to RBD, were no longer recommended. Examining the neutralization capacity of the mAbs against different recently emerged SARS-CoV-2 Omicron sub-lineage variants, compared to the earlier Delta variant, studies have shown that the majority of therapeutic mAbs lack the capacity to neutralize recently emerged SARS-CoV-2 Omicron sub-lineage variants [133,134,135] (Table 1). This indicates that, despite the many mutations that the RBD has undergone, these mutations were in non-conserved regions of the protein.

### 4.2. Vaccination against SARS-CoV-2

#### 4.2.1. Systemic Vaccines against SARS-CoV-2

The global epidemic and the increasing death toll due to COVID-19, along with the genome sequencing of SARS-CoV-2, accelerated the construction of vaccines targeting SARS-CoV-2 in an unprecedented, phenomenal fashion. The initial vaccinations against SARS-CoV-2 were introduced and commenced at the end of 2020 and aimed to elicit high levels of S protein-specific, particularly RBD-specific, antibodies [136]. The first European Medicines Agency (EMA)- and the American Food and Drug Administration (FDA)-approved vaccines were the Pfizer–BioNTech (vaccine: BNT162b) and Moderna (mRNA-1273) mRNA vaccines [137,138]. The currently approved parenteral vaccines are engineered based on different platforms, classified as mRNA, S protein/peptide subunit-based and viral vector-based vaccines (Table 2) [113,137,139,140,141].

The majority of the systemically administered vaccines are aimed at producing high levels of serum antibodies to diffuse into the respiratory mucosa, neutralize the virus, and prevent the disease [113]. Our group has demonstrated that, despite a greater rate of antibody decline, vaccinated individuals had considerably higher S protein-specific antibody levels than infected individuals [142]. Various studies have reported a rapid decline in patients’ humoral (neutralizing or anti-S protein antibody levels) and cellular response [143,144,145] after vaccination, which has been suggested to be linked to an increased susceptibility to SARS-CoV-2 infection [146]. Furthermore, in immunocompromised patients with solid tumors, multiple myeloma, or inflammatory bowel disease, we have demonstrated that booster vaccination with the mRNA COVID-19 vaccines reverses non-responsiveness and early antibody waning [147]. Although the systemic vaccinations were shown to elicit high levels of SARS-CoV-2-targeting nAbs, the levels stiffly waned and the vaccine effectiveness was reduced [148]. This, in association with the virus’ high mutational rate and the consequent evolvement of new variants, led to breakthrough infections [149], suggesting a better protection by monovalent vaccines targeting the respective variants of concern, as shown for the monovalent XBB1.5 vaccine [150]. 

To overcome the breakthrough infections caused by the emerged SARS-CoV-2 variants, different vaccination strategies were implemented. In 2022–2023, bivalent vaccines were designed and introduced to protect against both the original and the Omicron variants, BA.4/BA.5 of SARS-CoV-2 [151] and later XBB1.5. Vaccination with the bivalent vaccine has been shown to consist of antibodies not only specific to the S protein of the ancestral strain but also antibodies that cross-react to both variants’ S protein, suggesting the induction of a recall response to bivalent BA.5 vaccination, mainly to the shared epitopes on the S protein of the variants [152]. However, it was found that, in individuals who received the bivalent vaccine, the nAb levels to the Omicron variants were not significantly higher than in those receiving the monovalent vaccine, possibly due to a stronger effect of booster vaccination against the original strain [153]. 

Furthermore, vaccinations combining different COVID-19 vaccines, referred to as the ‘mix-and-match’ approach, also known as a heterologous boost COVID-19 vaccine strategy, have been practiced in the clinic and have demonstrated advantageous immunogenicity outcomes [154,155,156]. For instance, heterologous vaccination with the adenovirus-based ChAdOx1 (AstraZeneca) vaccine followed by an mRNA vaccine induced stronger immune responses compared to the homologous ChAdOx1 vaccine series [157,158]. In a recent single-blinded, randomized, parallel group superiority trial, the levels of SARS-CoV-2 neutralization antibodies and anti-RBD IgG levels were measured in participants who had received the first dose of the CoronaVac (inactivated SARS-CoV-2) vaccine followed by a dose of the BNT162b2 (mRNA-encoding S protein) or CoronaVac vaccine [159]. The results of this study indicated a significant increase in neutralizing antibodies following CoronaVac/BNT162b2 vaccination compared to the CoronaVac/CoronaVac regimen, further stressing the advantage of the mix-and-match vaccination [159]. In a phenomenon termed ‘antibody interference’, the presence of previously induced antibodies against SARS-CoV-2 may hinder the activity of antibodies introduced by mAbs or by vaccination at a later time point [160]. A suggested mechanism for this interference is that, for example, the presence of RBD-specific antibodies induced by the mRNA vaccines may hinder the therapeutic effect of mAbs that also target the same protein/epitope [160]. In a case study, however, a strong antibody response to RBD was observed after vaccination with two doses of the COVID-19 vaccine BNT162b2 within 40 days following COVID-19 mAb therapy [161].

Additionally, studies have shown that SARS-CoV-2 hybrid-immunity, referring to an immunity derived from infection and vaccination, in general, provides a more robust and durable protection [162,163,164]. This is attributed not only to a stronger induction of antibody responses, but also to qualitatively different T-cell responses due to exposure to antigens and immunodominant epitopes that are not included in the vaccine [158,165].vaccines-12-00795-t002_Table 2Table 2Approved parenteral COVID-19 vaccines, based on different categories.Vaccine Type/PlatformExpressed SARS-CoV-2 ComponentApproved COVID-19 VaccineDeveloperDosage Number and ScheduleReferenceNucleosideModified mRNA encoding S proteinCOMIRNATY(BNT162b2)BioNTech SE, Pfizer Inc.Two doses, 3 weeks apart[166]Modified mRNA encoding S proteinModerna vaccine(mRNA-1273)ModernaTwo doses, 4 weeks apart[167]Modifiedadenovirusvector Encoding S proteinVAXZEVRIA(ChAdOx1-nCoV-19)AstraZeneca,University of OxfordTwo doses given 4 to 12 weeks apart[168]Covishield(ChAdOx1 nCoV-19)Serum Institute of IndiaTwo doses given 12 weeks apart[169]Ad26CoV2.SJohnson & JohnsonOne time dose[170]CONVIDECIA(Ad5-nCoV)CanSino Biologics Inc.One time dose[171]Sputnik VGamaleya Research Institute of Epidemiology and MicrobiologyTwo doses given 3 weeks apart[172]InactivatedSARS-CoV-2BBIBP-CorVSinopharmTwo doses, 3 weeks apart[173]CoviVacRussian Academy of SciencesTwo doses given two weeks apart[174]CoronaVacSinovac Biotech Ltd.Two doses given 2 weeks apart[175]COVAXIN(BBV152)Bharat BiotechTwo doses given 4 weeks apart[176]VLA2001ValnevaTwo doses given 4 weeks apart[177]Adjuvantedprotein subunitDimeric RBD (with aluminium hydroxide)ZF2001Chinese Academy of SciencesThree doses given 30 days apart[178]Recombinant RBD fusion heterodimer of the Alpha and the Beta variants of SARS-CoV-2 (with an oil-in-water emulsion based on squalene (SQBA))PHH-1V (Bimervax)HIPRABooster dose (for 16 years and older age group)[179]PeptideSubunitA peptide vaccine composed of three short peptides derived from SARS-CoV-2 spike protein (S454–478, S1181–1202, and S1191–1211) conjugated to SARS-CoV-2 nucleocapsid proteinEpiVacCoronaVektor State Research Centre, RussiaTwo doses, 3-4 weeks apart[180]Recombinantprotein S protein nanoparticle NUVAXOVID(Nvx-CoV-2373)NovavaxTwo doses, 3 weeks apart[181]CovovaxSerum Institute of IndiaTwo doses, 3 weeks apart[182]


#### 4.2.2. Mucosal Vaccines against SARS-CoV-2

Like most respiratory viruses, SARS-CoV-2 can readily colonize the site of entry. Therefore, a vaccine that, in addition to systemic IgG, evokes protective mucosal responses mediated by s-IgA is more likely to limit transmission [23]. The mucosal vaccines that have received approval or are undergoing clinical trial evaluation are listed in Table 3. These vaccines use a range of delivery methods (nasal and oral dropper, sprayers (aerosolized), inhaler, and nebulized delivery) and vaccine platforms (DNA, RNA, protein-based, live-attenuated virus, and inactivated virus) (Table 3). As immunization based on only the mucosal route will elicit low and short-lasting immunity with minimal systemic protection, a prime-boost can be advantageous in achieving stronger mucosal and systemic protective responses.vaccines-12-00795-t003_Table 3Table 3Mucosal COVID-19 vaccines, approved or under clinical investigation.Vaccine Type/PlatformExpressed SARS-CoV-2 ComponentVaccine NameDeveloperPhaseClinical Trial IdentifierApplicationReferenceViral Vector (Replicating)live-attenuated influenza virus vector-basedexpressing SARS-CoV-2 RBDdNS1-RBDBeijing WantaiApprovedChiCTR2000037782 ChiCTR2000039715 ChiCTR2100048316Two intranasal doses, 14 or 21 days apart [183]Replication deficient Influenza A(CA4-DelNS1) virus expressing RBDdomain of S proteinDelNS1-2019-nCoV-RBDOPT1University of Hong Kong, XiamenUniversity and Beijing WantaiBiological PharmacyPhase 3ChiCTR2100051391Two intranasal doses, 14 days apart[184]Respiratory Syncytial virus expressingS proteinMV-014-212Meissa Vaccines, Inc.Phase 1NCT04798001Single intranasal dose, or 2 intranasal doses 36 days apart[185]Human adenovirus serotype 5expressing S protein and nucleocapsidhAd5-S-Fusion + N-ETSDImmunityBio Inc.Phase 1bNCT04591717Single subcutaneous dose followed by single sublingual dose,21 days apart [186]Adenoviral vector expressing WA1S proteinBBV154Bharat Biotech InternationalLimitedApprovedNCT05522335Two intranasal doses, 28 days apart[187]Viral Vector(Non-replicating)Parainfluenza virus 5 expressingWA1 S proteinCVXGA1/PIV5-SARS-CoV-2CyanVac LLCPhase 1NCT04954287Single intranasal dose[188]Adenoviral vector expressing S protein SC-Ad6-1Tetherex PharmaceuticalsCorporationPhase 1NCT04839042Single intranasal dose, or 2 intranasal doses one month apart[189]Adenoviral vector expressing S protein ChAdOx1/AZD1222University of Oxford andAstraZeneca BiopharmaceuticalsPhase 1NCT04816019Single intranasal dose[190]Adenoviral vector expressing WA1S proteinAd5-nCoV-IH(Convidecia Air)CanSinoBioApprovedNCT04552366Two doses with different administration routes (2 intranasal doses, 1 intramuscular and 1 intranasal dose, 28 days apart[191]Adenoviral vectorSputnik V/Gam-COVIDVacThe Gamaleya Research Institute of Epidemiology and MicrobiologyApprovedNCT04954092NCT05248373Single intranasal dose [192]Attenuated SARS-CoV-2 WA1 strainCoviLivCodagenix/Serum Institute ofIndiaPhase 3ISRCTN15779782Two intranasal doses, 28 days apart[193][194]Live attenuated virusRBD adjuvanted with aluminiumhydroxideCIGB-669(RBD + AgnHB)(Mambisa)Center for Genetic Engineeringand Biotechnology (CIGB)Phase 2RPCEC00000345Two intranasal dose, 28 days apart, orOne intramuscular dose followed by 2 intranasal doses, 28 days apart[195]Protein subunitS protein encapsulated by an artificial cellmembraneACM BiolabsACM-SARS-CoV-2- betaACM CpG vaccinecandidate (ACM-001)ACM BiolabsPhase 1NCT05385991Single dose, after full vaccination with any registered and commercial SARS-CoV-2 vaccines[196]Recombinant S proteinRAZI-COV PARSRazi Vaccine and Serum ResearchInstituteApprovedIRCT20201214049709N2Two intramuscular doses, followed by 1 intranasal dose[197]NCT, clinicaltrials.gov; ChiCTR, Chinese Clinical Trial Registry; IRCT, Iranian Registry of Clinical Trials; RPCEC, Cuban Public Registry of Clinical Trials.


## 5. Prime-Boost Vaccination against SARS-CoV-2

An approach termed ‘prime and spike regimen’ was investigated against SARS-CoV-2 [198]. The transgenic mice K18-hACE2, which express human ACE2, were intramuscularly administered with the Pfizer vaccine, and two weeks later the mice were administered intranasal with un-adjuvanted S protein. The study showed a robust systemic booster response that was comparable to parenteral administration of the Pfizer vaccine, associated with high levels of anti-SARS-CoV-2 IgA and IgG in the nasal wash and bronchoalveolar lavage (BAL) fluid. Also, mucosal T-cell immunity with the accumulation of S protein-specific CD8 T cells and antigen-experienced CD4 T cells in the lung and BAL fluid was induced. Furthermore, this vaccination strategy reduced transmission in a hamster model of SARS-CoV-2, and conferred protection against COVID-19-like disease after challenge with a lethal SARS-CoV-2 infection dose [198]. Based on this study’s results, the given persistent global COVID-19 infection rate, and the remarkable effectiveness of systemic vaccines in mitigating disease severity, a parenteral priming combined with intranasal boosting regimen may be the best vaccination strategy to protect against COVID-19 and also to prevent the transmission of SARS-CoV-2 (Figure 2).

In line with this approach, and by applying Syrian hamster as model of virus transmission, subcutaneous (parenteral) priming followed by intranasal boosting with spike HexaPro trimer formulated in a cationic liposomal adjuvant was shown to protect the animals against SARS-CoV-2 infection, suggesting an effective means to protect against the transmission of SARS-CoV-2 [199]. In a study testing a formulated human adenovirus serotype 5 expressing SARS-CoV-2 S and N proteins (hAd5 S-Fusion + N-ETSD), it was found that subcutaneous prime vaccination with an intranasal or subcutaneous boosting elicits greater T-cell responses than intranasal priming with a subcutaneous or intranasal booster [186]. Such a prime-boost immunization approach, mimicking hybrid-immunity, is being investigated in clinical settings (Table 3).

An important aspect for effective immunization and the success of vaccination, particularly in the prime-boost manner, lies in the presence of pre-existing immunity with the capacity to reduce the immunization effect [56]. With LAIV, it has been hypothesized that direct immunization with the vaccine in the respiratory mucosa is the mechanism for driving immune responses in the younger age groups that have low pre-existing viral exposure. As a result of minimal pre-existing or no immunity to influenza virus infections or exposures in infants and children, the vaccine’s type may serve as a mechanism for influencing the respiratory immunity [200]. The role of pre-existing antibodies in the context of SARS-CoV-2 has also been demonstrated [201]. A recent study, involving a cohort of patients primed with the mRNA-1273 or BNT162b2 vaccines, has shown that lower antibody levels prior to boost are associated with higher-fold increases in antibody levels following boosting, which suggests the role of pre-existing antibodies in modulating the immunogenicity of mRNA booster vaccines [201].

## 6. B-Cell Peptide/Mimotope-Based Vaccine

Considering the importance of prime-boost vaccination in establishing a strong immune response, a strategy to overcome the potential inhibition of neutralization by pre-existing immunity may be by the use of peptide-based vaccines that can target specific immunodominant regions of SARS-CoV-2 structural proteins. Unlike the robust antibody induction following COVID-19 mRNA-based vaccines, peptide-based vaccine modalities, based on using an adjuvant and thus allowing slow release (depot effect) of the antigen [202], would induce gradual induction of the peptide-specific antibodies, and such peptides may not be captured and neutralized by pre-existing antibodies which target the whole virus or protein. Furthermore, the vaccines in the prime-boost vaccination strategies that are currently being clinically investigated are either based on adenoviral vectors delivering S or N proteins of SARS-CoV-2 or on the adjuvanted/encapsulated/recombinant S protein or RBD of SARS-CoV-2 (Table 3). In addition to the pre-existing immunity against viral-based vaccines, as mentioned above, vaccination with whole proteins may induce the production of antibodies, which do not induce protection or confer a neutralizing capacity. Thus, vaccination with peptides representing the conserved regions on SARS-CoV-2 that are necessary for receptor binding and inducing neutralizing antibodies may be a more effective strategy.

The application of mAbs as therapeutic interventions binding to the SARS-CoV-2 RBD and potently neutralizing the virus has shown tremendous success in significantly reducing the severity of the diseases in infected patients at risk. However, mAbs-based therapies do not induce immunological memory and only offer the possibility of immediate protection in the case of exposure and infection with the virus. mAbs cannot substitute for vaccination, and are used for early treatment and post-exposure prophylaxis. However, the use of mAbs’ binding epitopes (mimotopes) has become a promising strategy both for infectious diseases and cancer [203,204,205,206]. Mimotopes [207] are peptides which mimic a natural binding B-cell epitope (or T-cell epitope) or the binding site of therapeutic mAbs, and, as such, they can be either linear or conformational epitopes [208,209]. Due to the homology between the mimotopes and their respective antigen epitope, they are capable of triggering an immune response [207,210]. Thus, mimotopes represent one of the suitable approaches toward the development of vaccines with better safety profiles [210,211,212,213,214].

In line with this approach, we have recently established a platform for the identification of linear and conformational B-cell epitope/mimotopes from therapeutic mAbs and their in vitro as well as in vivo characterization for the establishment of cancer vaccines [213,215]. Selected B-cell mimotopes are conjugated to a carrier protein, to use the peptides for immunization, and applied with an adjuvant to trigger both humoral and cellular responses [212,216,217,218]. An additional advantage of such a vaccination concept is that B-cell epitopes are HLA-independent, and, thus, no genetic preselection is necessary [212,219]. Active immunization with such B-cell mimotope-based vaccines enables the host to induce an epitope-specific response and the generation of antibodies with similar functionality to their respective mAbs [217]. Furthermore, the production of the antibodies is enhanced by the bystander stimulation of T-cells [212,220]. The B-cell peptides are conjugated to the carrier protein CRM197 (CRM, cross-reacting materials, an enzymatically inactive and nontoxic [toxoid] form of diphtheria toxin [221]) and administered in conjunction with the T helper 1 (Th1)/T helper 2 (Th2)-driving adjuvant Montanide, rapidly inducing Th1 and Th2 cells with a heterogeneous Th1 and Th2 cytokine profile for activating B cells [222] (Figure 3).

We have taken the approach of B-cell peptide-based vaccination against SARS-CoV-2 one step further by using it in a prime-boost vaccination approach to induce protective systemic and mucosal responses.

We applied our platform for the identification of conserved linear B-cell peptides/mimotopes as immunodominant epitopes on RBD. Three therapeutic SARS-CoV-2 mAbs, Sotrovimab, Cilgavimab, and Tixagevimab, which were clinically applied until recently (Table 1), were used to screen 15-mer linear overlapping peptides spanning the RBD of the original SARS-CoV-2 RBD. A unique immunodominant epitope was identified (Figure 4A). All examined mAbs were shown to bind to peptide #37, spanning the amino acids 444–458 and located within the RBM (Figure 4A,B) [96]. It has been shown that this region includes human T-cell epitopes with a capacity to induce a T-cell response in mice [223].

Following the identification of the immundominant epitope (mimotope #37, KVGGNYNYLYRLFRK), a vaccine compound consisting of the synthesized peptide, conjugated to CRM197 (P#37-CRM197), was used in a prime-boost immunization experiment (Appendix A). BALB/c mice were divided into three groups for intranasal immunization with P#37-CRM197, subcutaneous immunization with P#37-CRM197 in conjunction with the adjuvant Montanide, or subcutaneous priming (P#37-CRM197-Montranide) followed by intranasal boosting with (P#37-CRM197). 

Subcutaneous priming and intranasal boosting with the peptide led to higher levels of IgG and IgA antibodies in mice sera and BAL fluid samples (Figure 5A). To examine whether the prime-boost immunization of the mice also resulted in T-cell (cellular) responses, the cytokines interferon gamma (IFNg), IL-2, IL-4, IL-5, IL-10, and TNFa were evaluated in restimulated spleen cell cultures. As shown in Figure 5B, the levels of the cytokines were, in general, higher in the primed-boosted mice compared to those who underwent either subcutaneous or intranasal immunization alone.

The high IgG antibody levels indicated the systemic response induced by subcutaneous immunization. Additionally, in the subcutaneously immunized and prime-boosted mice, the production of the IgG subtypes IgG1 and IgG2a was higher in the mice sera and BAL fluid samples (Appendix A), suggesting a Th2 response leading to increased antibody production (IgG1) and a Th1 response leading to ADCC-mediating IgG2a antibodies.

The peptide identified and investigated in the above experiments represents a conserved region of the SARS-CoV-2 RBD [96], and pre-clinical experiments are currently ongoing to examine the peptide’s capacity to elicit antibodies which neutralize SARS-CoV-2 variants or other human coronaviruses compared to immunization with the full RBD-based vaccines. Furthermore, no data are yet available on whether the mucosal immune response induced by the peptide would confer protection from challenge with the whole virus. However, all in all, these results indicated that prime-boost immunization in mice with peptide #37, a conserved region in the RBM of S protein, has a capacity to induce strong humoral and cellular responses.

## 7. Conclusions

Sterilizing immunity, referring to the immune system’s capacity to eliminate a pathogen at the onset of the infection, i.e., directly at the site of entry and prior to the host cells’ infection, is the ultimate goal of interventions against SARS-CoV-2. However sterilizing immunity against SARS-CoV-2 is hard to achieve due to waning immunity and viral antigenic evolution [224,225]. Even though SARS-CoV-2 initially infects the upper respiratory tract, leading to the first interactions with the immune system, vaccination against COVID-19 has, until recently, been largely focused on inducing strong systemic responses and the production of serum nAbs, and targeting the virus by a mucosal vaccine has only recently started gaining traction. So far, only two mucosal vaccines against SARS-CoV-2 have been approved, and more groups, including us, are investigating mucosal immunity against SARS-CoV-2, aiming to construct a mucosal vaccine against the virus. As also presented in this review, the superiority of mucosal vaccination against SARS-CoV-2, compared to only systemic vaccination, can be further enhanced by combining systemic priming and intranasal boosting, leading to a stronger mucosal and systemic response than either route alone. Therefore, it is not unlikely that, in the years to come, an increasing number of mucosal vaccines against SARS-CoV-2 will be approved for use worldwide and, possibly, all the COVID-19 vaccines will become mucosal-based.

## Figures and Tables

**Figure 1 vaccines-12-00795-f001:**
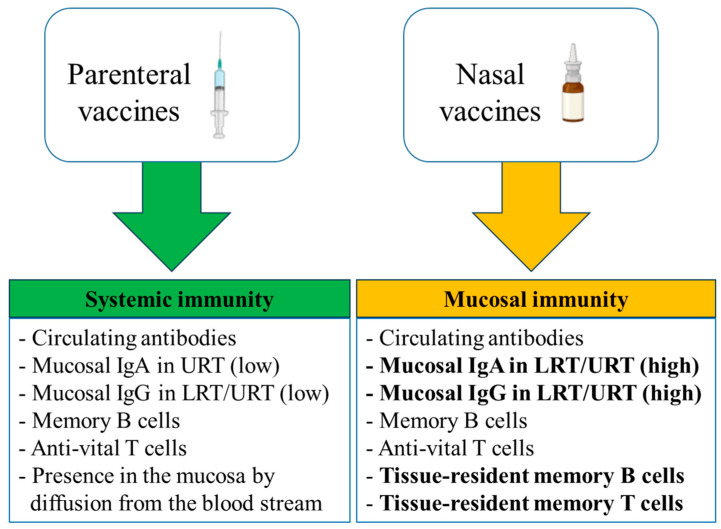
Comparison of parenteral and mucosal vaccination outcomes. Parenteral vaccination induces substantial systemic responses, although with low levels of mucosal IgA and IgG antibodies in the LRT and URT, which are induced at high levels following mucosal vaccination. The tissue-resident memory B and T cells provide antiviral environment at the time of infection and prevent transmission. The main specific characteristics of the mucosal immunity are shown as bold text.

**Figure 2 vaccines-12-00795-f002:**
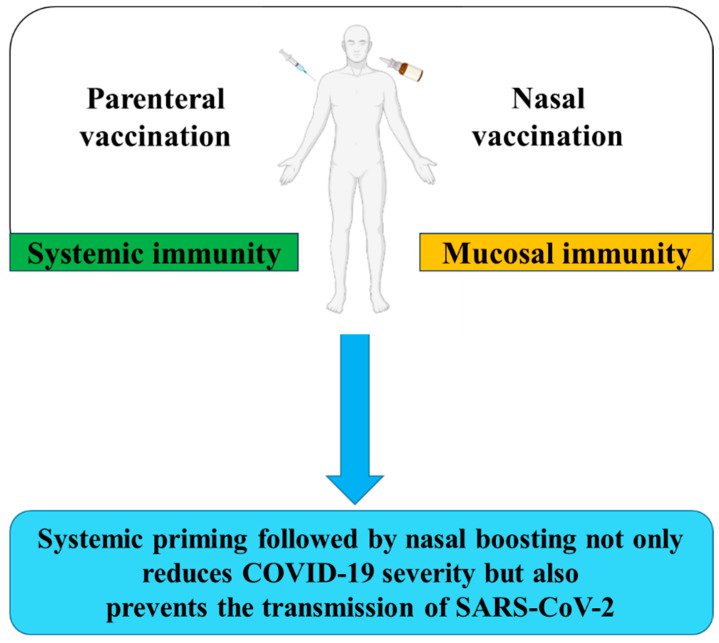
Combining systemic priming and nasal boosting can potentially be a more effective approach to COVID-19 (Li et al., 2022; Katsande et al., 2022 [199,200]).

**Figure 3 vaccines-12-00795-f003:**
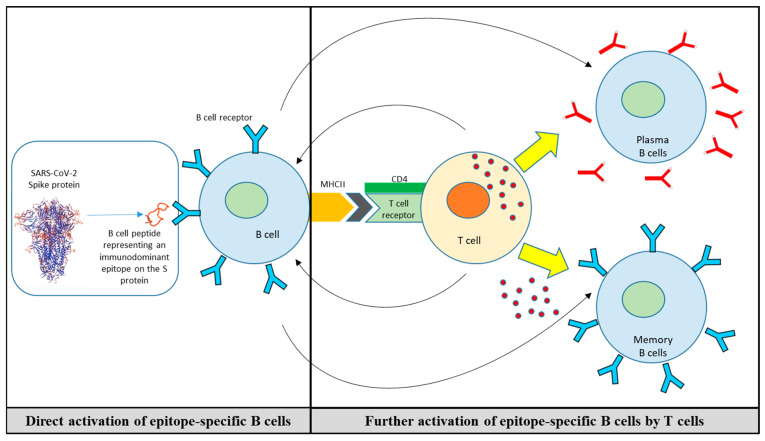
Activation of B cells and T cells by B-cell peptide-based vaccine. The constructed B-cell peptide-based vaccine not only directly activates the B cells for production of antibodies but, due to the T-cell epitopes in the carrier protein conjugated to the B-cell peptide, also activates CD4 T cells and T follicular helper (TFH) cells that further enhance the activation and affinity maturation of B cells.

**Figure 4 vaccines-12-00795-f004:**
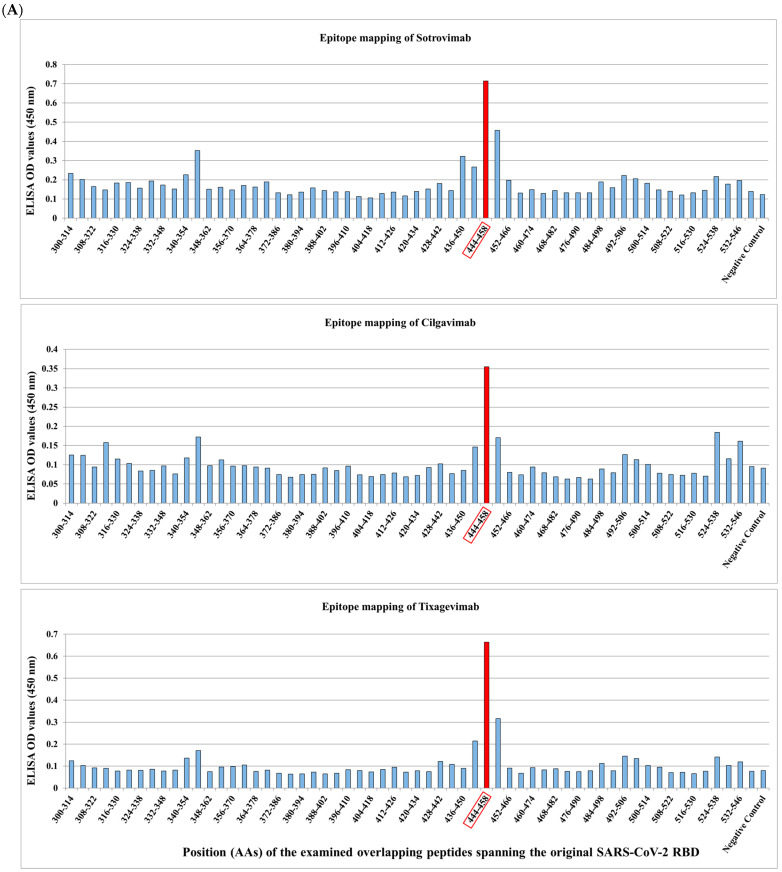
(**A**) Peptide enzyme-linked immunosorbent assay (ELISA) for binding evaluation of Sotrovimab, Tixagevimab, and Cilgavimab (5µg/mL) to 15-mer biotinylated overlapping peptides spanning the SARS-CoV-2 RBD, as described in Appendix A. Highlighted in orange is the dominant peptide 37. (**B**) The position of the identified peptide #37 is shown on the S protein trimer (side view), generated in PyMOL (PyMOL Molecular Graphics System, Version 2.5.0a0, Schrödinger, LLC).

**Figure 5 vaccines-12-00795-f005:**
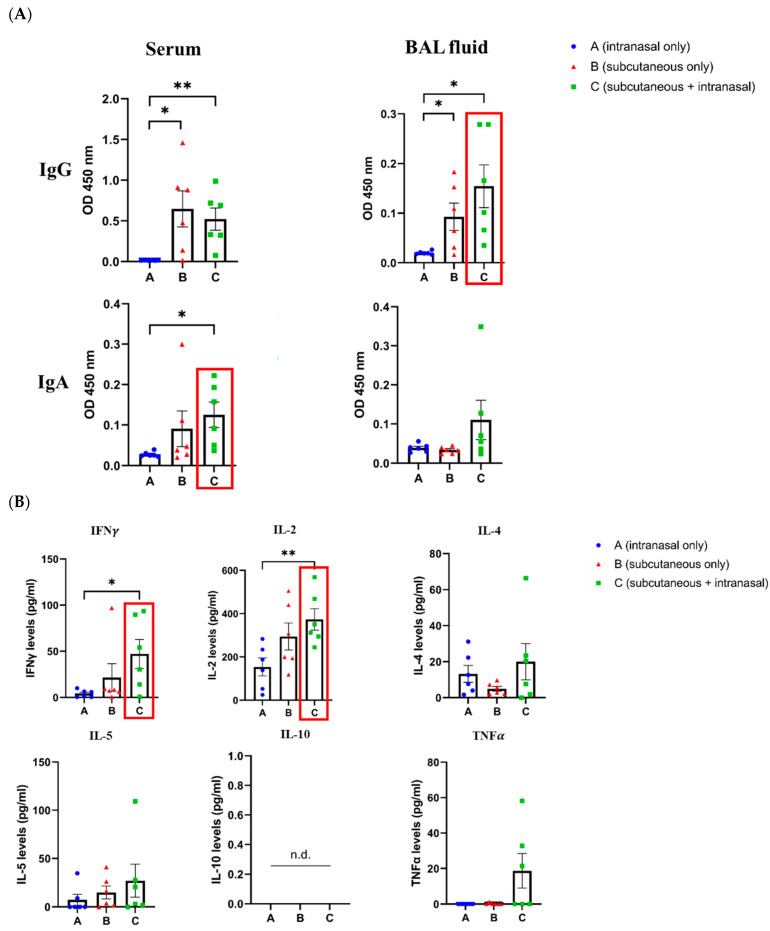
Humoral and cellular responses in mice immunized either subcutaneously or intranasally, or with a combination of both (prime-boost). The levels of peptide-specific IgG and IgA antibodies were assessed by ELISA, using the sera or BAL fluids of mice from all experimental groups (**A**), and T-cell (cellular) responses were evaluated based on the levels of secreted cytokines in restimulated mice splenocytes (**B**) Appendix A. Significant differences are indicated by asterisks (* = *p* < 0.05, ** = *p* < 0.01) in each graph.

**Table 1 vaccines-12-00795-t001:** Selected therapeutic/prophylactic mAbs—characteristics and activity against different variants of SARS-CoV-2.

Name	Date of Approval	Treatment	Targeting Epitope	Administration Route	Neutralization Activity against the SARS-CoV-2 Variant Delta (EC_50_, ng/mL) [135] *	Fold-Reduction of Neutrlization Activity (EC_50_, ng/mL) against Delta BA.2, BA.5, BA.2.75.2, XBB, BQ.1, and BQ.1.1 Variants (Fold-Reduction) [135] *					
						BA.2	BA.2.75.2	BA.5	BQ.1	BQ.1.1	XBB
Bebtelovimab	February 2022	Therapeutic	RBD	Intravenous	0.4						
Sotrovimab	May 2021	Therapeutic	RBD	Intravenous	98.6						
Casirivimab	November 2020	Therapeutic Or Post-exposure prophylaxis	RBM	Intravenous or sub-cutaneous	14.7						
Imdevimab					20.1						
Cilgavimab	December 2021	Pre-exposure prophylaxis	RBD	Intramuscular	21.7						
Tixagevimab					12.4						

* Adapted based on Touret et al., 2023 [135]. EC50: half-maximal effective concentration (ng/mL). RBD: receptor-binding domain, RBM: receptor-binding motif. The fold-reduction levels are depicted in the following color codes: 1–5 fold: **▬** 6–10 fold: **▬** 11–15 fold: **▬** 16–20 fold: **▬** Non-neutralizing: **▬**.

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
