# Peer review of "SARS-CoV-2 Vaccines: The Advantage of Mucosal Vaccine Delivery and Local Immunity"

_vaccines, 2024, doi:10.3390/vaccines12070795_

Round 1

Reviewer 1 Report

Comments and Suggestions for Authors

General Comments

The paper "Why Should Vaccines against Respiratory Diseases Go Mucosal? B Cell Epitope-Based Vaccines against SARS-CoV-2" by Tobias  et al. begins by stating in the title a general goal of the article Why Should Vaccines against Respiratory Diseases Go Mucosal and the focuses immediately onon  SARS-CoV-2 on SARS-CoV-2 and then ends with a sharper specific focus on the authors' research interest B cell epitope vaccines.

This is both a confusing and misleading task imposed on the reader who may be challenged of what to expect from the title . The authors should consider  changing the title to something like "SARS-CoV-2  vaccines why the mucosal route of delivery is important"

Specific Comments

line 65-69, These sentences are not clear an are stated poorly...a better way may be...""In serum, IgA is present in a monomeric form, while mucosal secretory IgA (s-IgA) is in a polymeric form, particularly dimeric or tetrameric [8,9]. Mucosal IgA differs from serum IgA in its molecular structure. Serum IgA consists of monomeric molecules with a 7S structure. Mucosal IgA, on the other hand, includes an additional component, the secretory component, which gives it a larger 11S structure. This structural difference prevents mucosal IgA from binding to the Fcα-receptor I (FcαRI) expressed by myeloid cells, such as monocytes, neutrophils, and some subsets of macrophages and dendritic cells (DCs), thereby avoiding the induction of pro-inflammatory responses [10]."

Table 1 is confusing and should be eliminated

Ufortunately some of the seminal studies describing the discovery of mucsal IgA responses to viral in fection and papers describing differences in mucosal and antibody responses to live and killed vaccines are missing 

The authors mifht consider inclusion some of these. 

1.     Artenstein MS, Bellanti JA, Buescher El. Identification of The Antiviral Substances In Nasal Secretions. Proc Soc Exp Biol Med. 1964 Nov; 117:558-64.

2.     Bellanti JA, Artenstein MS, Buescher EL. Characterization of Virus Neutralizing Antibodies In Human Serum And Nasal Secretions. J Immunol. 1965 Mar; 94:344-51.

3.     Bellanti JA, Sanga RL, Klutinis B, Brandt B, Artenstein MS. Antibody responses in serum and nasal secretions of children immunized with inactivated and attenuated measles-virus vaccines. N Engl J Med. 1969 Mar 20;280(12):628-33.

4.     Bellanti JA, Artenstein MS, Brandt BL, Klutinis BS, Buescher EL. Immunoglobulin responses in serum and nasal secretions after natural adenovirus infections. J Immunol. 1969 Nov;103(5):891-8.

5.     Ogra PL, Karzon DT, Righthand F, MacGillivray M. Immunoglobulin response inserum and secretions after immunization with live and inactivated poliovaccine and natural infection. N Engl J Med. 1968 Oct 24;279(17):893-900.

6.     Ogra PL, Kerr-Grant D, Umana G, Dzierba J, Weintraub D. Antibody response inserum and nasopharynx after naturally acquired and vaccine-induced infectionwith rubella virus. N Engl J Med. 1971 Dec 9;285(24):1333-9

7.     Bellanti JA, Zeligs BJ, Mendez-Inocencio J, et al. Immunologic studies of specific mucosal and systemic immune responses in Mexican school children after booster aerosol or subcutaneous immunization with measles vaccine. Vaccine. 2004 Mar 12;22(9-10):1214-20.

Reviewer 2 Report

Comments and Suggestions for Authors

This article systematically provides a comparison between parenteral and mucosal vaccination outcomes for respiratory disease, using SARS-CoV-2 as an example. The discussion is highly topical and relevant, therefore will be of high interest for the research community.

The value of this paper is that it provides a comprehensive knowledge base for the SARS-CoV-2 viral biology, current understanding of the immune responses, and pros and cons of currently available prophylactics. It includes several tables that summarise different COVID-19 vaccines in current development. It went on to discuss different vaccine regimes currently in the development pipeline, with evidence that builds a strong case for the “parenteral prime- mucosal boost” vaccination strategy advocated by the group.

Overall this is an excellent manuscript with a wealth of useful information.

Some minor suggestions below:

Line 281: It will be better to make it clear that the rapid decline in immune responses refers to the post-immunisation scenario.

Section 4.2.2: suggest revising the first sentence. For example “Like most respiratory viruses, SARS-CoV-2 can readily colonize the site of entry”

Page 16, line 52: it isn’t clear why “circumvent the neutralizing effect of pre-existing antibodies” is a benefit.

Figure 4A may be replaced by a more effective format. The current figure can be supplied in supplementary materials as the font is too small to be readable.

Section 6: can you provide some more information/comparison between the mimotope vs linear B  cell epitope?

Apart from breakthrough infections and reinfections, perhaps the discussion can also touch on the non-symptomatic infections where the carriage of virus in the mucosa serves as viral reservoir and posts risks for the more vulnerable individuals. This is another place where mucosal immunity can exert some protective effects.

Round 2

Reviewer 1 Report

Comments and Suggestions for Authors

Authors have satisfactorily responded to to all queries.